# “Fingerprinting” Benign and Cancerous Skin Lesions Using Vibrational Optical Coherence Tomography: Differentiation among Cancerous Lesion Types Based on the Presence of New Cells, Blood Vessels, and Fibrosis

**DOI:** 10.3390/biom12101332

**Published:** 2022-09-21

**Authors:** Frederick H. Silver, Tanmay Deshmukh, Nicole Ryan, Arielle Romm, Hari Nadiminti

**Affiliations:** 1Department of Pathology and Laboratory Medicine, Robert Wood Johnson Medical School, Rutgers, the State University of New Jersey, New Brunswick, NJ 08854, USA; 2OptoVibronex, LLC., Allentown, PA 18104, USA; 3Summit Health, Dermatology Department, Berkeley Heights, NJ 07922, USA

**Keywords:** collagen, fibrous tissue, stroma, actinic keratosis, basal cell carcinoma, squamous cell carcinoma, melanoma, skin cancer

## Abstract

In this study, we use vibrational optical coherence tomography (VOCT) to examine the morphology and stiffness of benign and cancerous lesions. Lesion images and 3D plots of weighted displacement versus frequency and depth were used to compare the cellular, dermal collagen, new blood vessels, and fibrotic composition of normal skin, actinic keratoses (AK), nodular and superficial basal cell carcinomas (BCCs), squamous cell carcinomas (SCCs), and melanomas. The results of this study suggest that benign and cancerous lesions differ based on the addition of new cells with increased resonant frequency and stiffness (80 Hz, 1.8 MPa), new blood vessel peaks (130 Hz, 4.10 MPa) that appear to be less stiff than normal blood vessels, and new fibrous tissue peaks (260 Hz, 15–17 MPa) that are present in carcinomas but not in normal skin and only partially present (80 Hz and 130 Hz only) in AKs. Results obtained by creating images based on the location of the 80 Hz, 130 Hz, and 260 Hz peaks of cancerous skin lesions suggest that the fibrous tissue appears to surround the new cells and new lesion blood vessels. The results of this study suggest that the morphology and location of the fibrous tissues in relation to the new cancer-associated cells and lesion blood vessels may provide information on the invasiveness and metastatic potential of skin cancers. The invasiveness and metastatic potential of melanomas may be a result of the cancer-associated cells laying down fibrous tissue that is used as a pathway for migration. The new cancer-associated blood vessels in the vicinity of the new cancer-associated cells may promote this migration and eventual metastasis. The ratios of peak heights 50/130 Hz and 80/130 Hz of normal cells, new lesion cells, new lesion blood vessels, and fibrotic tissue may be used as a “fingerprint” for detecting melanoma and to differentiate it from other skin cancers non-invasively using VOCT.

## 1. Introduction

Organ fibrosis affects all major human tissues and can lead to the excessive accumulation of extracellular matrix (ECM) components, predominantly collagens. It can lead to compromised organ function and failure and is estimated to contribute to 45% of all deaths in the developed world [1]. Numerous clinical and pathological observations have established a clear relationship between chronic inflammation, fibrosis, and cancer dating back almost 200 years [2]. Evidence reported in the literature suggests that cancer-associated fibroblasts (CAFs), but not normal fibroblasts, support metastatic lesions [3]. Recent evidence that CAFs and tumor cells are regulated by provisional matrix molecules, leading to continued stromal collagen cross-linking, which results in fibrosis [4]. Cancer-associated fibroblasts (CAFs) are major components of the surrounding stroma of carcinomas. They emerge in the tumor microenvironment because of signals derived from the cancer cells. Biochemical crosstalk between cancer cells and CAFs as well as mechanical remodeling of the stromal extracellular matrix (ECM) by CAFs are important contributors to tumor cell migration and invasion. These steps are critical for the progression of a primary tumor leading to metastatic disease [5].

The secretion of lysyl oxidase (LOX) by tumor cells initiates lysine-derived collagen cross-linking, thereby stiffening the tumor stroma. This creates mechanical forces that trigger the integrin-mediated formation of focal adhesions that initiate tumor cell invasion [6]. LOX-mediated collagen crosslinking is responsible for fibrosis-enhanced metastasis [6,7]. It also directly increases tumor cell proliferation, enhancing metastatic colonization and growth [6], and leads to increased metastasis [6]. Breast cancer cells recruit surrounding stromal cells and CAFs to remodel their extracellular matrix (ECM) and promote invasive tumor growth [6,7].

Two major extracellular matrix (ECM) components, fibronectin (F_n_) and collagen I (Col I), are known to interact with each other to regulate cellular behavior. ECM remodeling is primarily mediated by matrix metalloproteinases (MMPs) [8] and transforming growth factor (TGF)-β–integrin signaling is implicated in normal physiology. ECM remodeling is also involved in a variety of pathological processes, including systemic sclerosis, idiopathic pulmonary fibrosis, chronic obstructive pulmonary disease, and cancer [9,10,11,12]. Collectively, these findings provide the mechanisms by which F_n_ and Col I synergistically interplay in promoting a sustained altered signaling cascade to remodel the breast tumor stroma for invasive breast tumor growth [9].

Cancer cells are unable to assemble their own F_n_ matrix; surrounding cells in the stroma of CAFs are reported to up-regulate F_n_ [10]. These tumor-secreted factors mediate the early stromal assembly of a thick, dense, and stretched F_n_ matrix comprising highly unfolded molecules [11]. This altered F_n_ matrix not only exhibits increased stiffness and viscosity, but also mediates enhanced pro-angiogenic signaling. The establishment of a fibrotic-like microenvironment via TGFβ-driven collagen I production may generate a fertile soil around dormant tumor cells that then drives the transition from dormancy to metastatic growth through a β1-integrin signaling mechanism [8,12].

Both cancer cells and epithelial cell-derived cancerous tissue are stiffer than normal tissue [13,14,15,16]. The increased stiffness of cancer cells and fibrous tissue found in skin carcinomas has been studied using a new non-invasive technique termed vibrational optical coherence tomography [17,18,19,20,21,22,23,24]. The technique applies audible sound and infrared light transversely to the surface of skin that provides an image and a measurement of the resonant frequency and stiffness of the major skin components [17,18,19,20,21,22,23,24]. Normal skin has major resonant frequency peaks at 50 Hz (stiffness about 1.2 MPa), 100 Hz (2.5 MPa), 150 Hz (4.5 MPa), whereas actinic keratoses have new peaks at 80 Hz (1.8 MPa) and 130 Hz (4.1 MPa), and basal cell carcinomas, squamous cell carcinomas, and melanomas have new peaks at 80 Hz, 130 Hz, and 260 Hz [17,18,19,20,21]. The 50 Hz (1.2 MPa) peak reflects the behavior of epidermal cells and fibroblasts, whereas the 150 Hz (4.5 MPa) peak is seen in normal arteries and veins [24]. The new peaks at 80 Hz (1.8 MPa) and 130 Hz (4.1 MPa) have been hypothesized to reflect the presence of new pre-cancerous cells and perhaps CAFs that are associated with new friable blood vessels (130 Hz peak). The 260 Hz peak observed in all cancerous skin lesions is associated with the formation of fibrous tissue with moduli between 15 and 17 MPa, which is much higher than the modulus of dermal collagen (2.5 MPa) [17,18,19,20,21]. The elastic moduli measured using vibrational optical coherence tomography (VOCT) are higher than moduli reported using elastography, since VOCT provides a transverse surface deformation that relays information about the component tensile elastic moduli (in-phase deformation with sinusoidal sound wave) as opposed to the shear properties of the interfibrillar matrix obtained through elastography, as discussed previously [22].

The purpose of this study was to provide non-invasive 3D maps of the location, stiffness, and morphology of the fibrous tissue and new cells deposited in cancerous skin lesions. This information is used to evaluate the relative relationship between the location of 80 Hz cells, 130 Hz blood vessels, and the location and orientation of the fibrotic tissue at 260 Hz that is deposited near each type of skin cancer. Preliminary results suggest that the geometric relationship, resonant frequency peak location, and relative peak heights of new cells, new lesion blood vessels, and fibrous tissue, as well as the ratio of the major resonant frequency peak heights, may provide information that can be used to “fingerprint” and predict the invasive and metastatic behavior of different skin cancers.

## 2. Materials and Methods

### 2.1. Subjects

Normal skin from 14 subjects (11 males and 3 females) was studied in vivo using VOCT after informed consent was obtained, as reported previously [18,19]. Figure 1 is a diagram showing the setup for both in vivo and in vitro experiments conducted in this study. The data were collected in the reflectance mode in both experiments after the application of a sinusoidal sound wave at 55 dB SPL. Control skin was examined from the hands, arms, and legs of each subject. The resonant frequencies of the components of the skin were measured in vivo by mounting the OCT hand piece on a custom-built universal mount that was supported over the area of skin to be studied [18,19]. The subjects studied ranged in age from 21 to 71 years old.

Suspicious skin lesions identified by dermoscopy in the Dermatology Clinic at Summit Health (Westfield, NJ, USA) were biopsied and studied in vitro using VOCT on a sample microscope stand, as shown in Figure 1 [18,19]. These lesions appeared during dermoscopic examination to contain layers of cells, superficial blood vessels, or abnormal color. They were obtained from the arms, back, legs, abdomen, and necks of subjects at Summit Health. The biopsied lesions were studied blindly by VOCT without identification of the age and sex of the patient. Fifty-eight biopsies of complete excisions and fifty-one Mohs sections were examined using VOCT as part of an IRB-approved study. Several of the biopsies were large enough to make multiple measurements on different areas of each sample. VOCT measurements were made on areas of about 0.0625 mm^2^. All subjects signed consent forms prior to enrolling in the study.

### 2.2. OCT Images and Scans of Pixel Intensity Versus Depth

OCT image collection was accomplished using a Lumedica Spectral Domain OQ 2.0 Labscope (Lumedica Inc., Durham, NC, USA) operating in the scanning mode at a wavelength of 840 nm. The device generates a 512 × 512-pixel image with a transverse resolution of 15 μm and an A-scan rate of 13,000/sec. The beam size used was 0.25 mm in diameter, as discussed previously [18,19]. All grayscale OCT images were color-coded to enhance the image details. For the pixel intensity versus depth plots, the surface of the sample was electronically traced, and the averages of pixel values were calculated along the surface of the sample, which were then plotted against the depth. For curved biopsy specimens, the tracing was performed parallel to the surface of the image.

Multiple cross-sectional OCT scans were collected using the volume scan software included in the Lumedica OQ Labscope and processed into 3D images using Image J software.

The pixel intensities obtained from the grayscale images were plotted versus depth for each sample studied. The enhanced OCT images used darker-colored (blue and purple) regions to reflect lower pixel intensities, and the lighter (yellowish) regions reflected higher pixel intensity regions. Pixel intensities were processed using Image J software, analyzed with a MATLAB program, and plotted versus skin depth. Previous studies have shown that the images of normal skin and cancerous lesions seen by OCT correlate with the histological images seen in sections cut from tissue biopsies [17,20,21].

Maps of different skin anatomical structures were created by examining slices collected using the OCT volume scan app as well as by looking at changes in the pixel intensity vs. depth scans of the image. These data were combined with the measurement of the weighted displacement versus frequency data to create a 3D map of the components of each lesion [18,19]. These data were used to map the location of cells, new blood vessels, and fibrous tissue in each type of lesion.

### 2.3. Measurement of Resonant Frequency and the Elastic Modulus

The OQ Labscope 2.0 was modified by adding a 2 inch-diameter speaker placed about 2.5 inches from the tissue in the VOCT studies [18,19]. A sinusoidal sound wave at 55 dB was produced using an app supplied by the I5 computer that is part of the Labscope. Both the sound wave and the light were applied perpendicular to the surface of the sample along the axis of the light beam. The deflection of the surface along the direction of the light beam was measured. All weighted displacement measurements were made from A line data at a single point based on the location of the lesion using the OCT lesion image [18,19]. All lesions showed up as highly dense black spots in the OCT image. The Labscope was also modified to collect and store single raw image data that were used to calculate sample displacements (amplitude information) from A line data. The data were processed using MATLAB software, as discussed previously [18,19]. The displacement of the tissue was detected by measuring the frequency dependence of the deformation based on the reflected infrared light and filtered to collect only vibrations that were in phase (elastic component) with the sound input signal. All samples were pulsed five times with the sinusoidal sound wave before the data were collected at each frequency. The sample was allowed to recover for 3 sec before the sound frequency was increased. Vibrations for each frequency were isolated to calculate the amplitude. These amplitudes were plotted against the frequency of the vibrations. The result is a spectrum of displacements for specific tissue components as a function of the frequency of the applied sound; the resonant frequency of each tissue component, e.g., cells (50–80 Hz), dermal collagen (100–120 Hz), blood vessels (130–150 Hz), and fibrotic tissue (180–260 Hz), have been assigned previously based on studies on a variety of soft tissues and polymeric materials [18,19,20,21,22]. Maps of the location of the new cells (80 Hz peak), new lesion blood vessels (130 Hz), and fibrous tissue (280 Hz peak) were constructed from measurements of resonant frequency, sample thickness, pixel intensity versus depth scans, and volume scans of the tissue.

The resonant frequency of a tissue component is defined as the frequency at which the maximum in-phase displacement is observed in the amplitude data for that component. The measured resonant frequencies are converted into elastic modulus values using a calibration equation (Equation (1)) based on in vitro uniaxial mechanical tensile testing and VOCT measurements made on the same tissue at the same time, as reported previously [23,24]. The resonant frequency of each sample is determined by measuring the displacement of the tissue resulting from transversely applied sinusoidal audible sound-driving frequencies ranging from 30 Hz to 300 Hz, in steps of 10 Hz. This frequency range excludes vibrations from underlying nerves, muscles, and bones, as reported previously [24]. The peak frequency (the resonant frequency), fn, is defined as the frequency at which the displacement is maximized. The peak heights both in the presence and absence of the sample are measured, and normalization is achieved by dividing the measured sample peak displacements by the displacements measured in the absence of the sample. This yields a normalized weighted displacement value, as discussed previously [22,23,24]. Sample component displacements are inversely related to the modulus of the tissue elements.
(1)$$E*d=0.0651*{\left(\mathrm{fn}\right)}^{2}+233.16$$

Calibration studies using in vitro uniaxial tensile testing and VOCT measurements were used to develop an Equation (1) for soft tissues. Since soft tissues have a density very close to 1.0, Equation (1) is valid for most tissues found in the body; where the thickness d is in m and is determined from OCT images, fn^2^ is the square of the resonant frequency, and E is the elastic modulus in MPa, as discussed previously [23,24]. Equation (1) was used to calculate the modulus values listed in Table 1.

Normal skin studies were conducted in vivo using a universal hand piece mount that was constructed to hold the OCT handpiece (Figure 1). Tissue biopsies were studied in vitro by VOCT using the microscope stage within 5 min of harvesting by the dermatologist and kept wet using moist saline-impregnated gauze during testing (Figure 1). Previous studies on skin lesions in vitro and in vivo indicate that the resonant frequency of cells, new blood vessels, and fibrous tissue are identical [17,18,19,20,21,22,23,24]

Once VOCT studies were conducted, the biopsy samples were immersed in fixative and transported to the pathology lab for diagnosis. Histopathology on skin biopsies was conducted by a dermatopathologist after routine dehydration in alcoholic solutions, embedding in paraffin, thin sectioning, and staining with H&E. Mohs thin sections were processed after fixation by frozen sectioning and H&E staining. They were reviewed by a trained Mohs dermatopathologist who conducted the pathological analysis.

## 3. Results

Results of the VOCT experiments conducted on normal skin in vivo and excisions of AK, BCC, SCC, and melanoma illustrate that new resonant frequency peaks are seen in skin lesions. Note that the new cellular (80 Hz) and fibrotic peaks (260 Hz) seen in cancerous skin lesions are stiffer than those found in normal skin (50 Hz and 100 Hz). BCC, SCC, and melanoma all have peaks at 80, 130, and 260 Hz, indicating that there appears to be a relationship between the presence of new cells at (80 Hz), new lesion blood vessels (130 Hz), and fibrous tissue (260 Hz) in cancerous lesions. Previous published results indicate that the resonant frequency peaks seen in skin and skin lesions in vitro are like those observed in vivo [17,18,19,20,21,22,23,24]. Table 1 lists the results of resonant frequency measurements and elastic modulus values obtained in the VOCT studies conducted on normal skin, AK, BBC, SCC, and melanoma specimen. Note that since the resonant frequency is measured at 10 Hz intervals, there are some variations in the resonant frequency seen from sample to sample. Figure 2 shows a 3D color-coded OCT image of normal skin (A) from a 71-year-old subject illustrating the location of dermal cells and papillary collagen. Figure 2A also shows a visual image of the normal skin as well as an OCT image (Figure 2B) of the area studied by VOCT. Figure 2C is a plot of weighted displacement versus frequency and versus depth showing the location of skin cells, papillary collagen, and blood vessels. Top and bottom views of the location of the cells and dermal collagen in normal skin where VOCT measurements were made are also shown in Figure 2A. Note that the peaks at about 50 Hz (cells), 100 Hz (dermal collagen), and 150 Hz (blood vessels) are typical of what is found in normal skin. The cells are found primarily in the epidermis and papillary dermis, and the dermal collagen and blood vessels are found in the papillary dermis based on the data shown in Figure 2C.

Figure 3A shows a 3D reconstruction of a typical actinic keratosis (AK) illustrating the top and bottom views of the lesion (left: the location of new cells and blood vessels colored in red) as well as a camera view of the lesion (center). Figure 3B shows the OCT of the cross-section of the lesion located at the arrow. The figure illustrates the presence of a new cellular peak at 80 Hz (shown in red in 3A). The locations of these new peaks are shown in Figure 3C. Normal skin has a cellular peak at 50 Hz, a papillary dermal collagen peak at 100 Hz, and a blood vessel peak at 150 Hz. Note the peaks at 80 Hz and 130 Hz seen in AK that are not seen in normal skin. The organization of the new cells and blood vessels in AK is in an almost linear fashion in Figure 3A.

Figure 4A shows a 3D color-coded CT image of a nodular basal cell carcinoma (BCC) on the left (A) and a camera image of the lesion on the right. Note the location of the new lesion blood vessels and cells in red and the location of the new fibrous tissue in yellow in A. Figure 4B shows an OCT image of the cross-section where the data were collected. Note that the lesion is seen as a very small black spot in B below left arrow. Figure 4C shows a 3D plot of weighted displacement versus frequency and versus depth showing the location of skin cells (50 Hz), a new cellular peak (80 Hz), dermal collagen (100 Hz), new lesion blood vessels (130 Hz), and fibrotic tissue (260 Hz). Note that the localization of the new cellular and the new blood vessel peaks in A are in lesion islands separated from other lesion islands. The fibrous tissue appears to surround the new cellular material and new blood vessels.

Like in the nodular BCC, the new cellular peak (80 Hz) and new blood vessel (130 Hz) peak of a superficial BCC (Figure 5) looks like the peaks of the nodular BCC shown in Figure 4. The location of the new cellular peak (80 Hz), new blood vessel peak (130 Hz), and fibrotic tissue (260 Hz) are like the nodular BCC with additional orientation along the surface of the lesion (Figure 5C). Note the formation of cellular islands in both nodular and superficial BCCs.

Figure 6 shows color-coded OCT images of a squamous cell carcinoma (SCC). Figure 6A (left) illustrates the location of the new cells (80 Hz) and new blood vessels (130 Hz) in red, and fibrotic tissue (yellow). The camera image is shown in the center along with arrows pointing to where the data were collected. Figure 6B is an OCT cross-section of superficial BCC. Figure 6C shows a 3D plot of weighted displacement versus frequency and versus depth, showing the location of skin cells (50 Hz), new cells (80 Hz), papillary collagen (100 Hz), new lesion blood vessels (130 Hz), and fibrotic tissue (260 Hz). Note that the relationship between new cells and fibrotic tissue appears to occur in both islands like that observed with BCC (Figure 4 and Figure 5) and like that observed with melanoma (see Figure 7A).

Three-dimensional color-coded images of a melanoma are shown in Figure 7A (left) along with a camera image of the location in the lesion where the VOCT data were collected (center). The lesion in the OCT image of the cross-section (Figure 3B) where the data were collected is the linear blue area in the center of the lesion near the top. Figure 7C shows a 3D plot of weighted displacement versus frequency and versus depth, showing the location of the new cellular peak, new lesion blood vessel peak (130 Hz), and new fibrotic peak (260 Hz). Note that the fibrotic tissue peak (260 Hz) is small in the melanoma probably due to the absorption of infrared light by the melanin in the lesion.

Figure 2 through Figure 7 illustrate that the height of the 100 Hz peak dominates the resonant frequency spectrum of normal skin, whereas the 80 Hz and 130 Hz peaks dominate the spectrum for AK, the 260 Hz peak dominates for BCC, the 50 Hz and 260 Hz peaks dominate for SCC, and the 80 Hz and 260 Hz peaks dominate for melanoma. By normalizing the average peak heights in each spectrum by the largest resonant frequency peak in each sample type (Figure 8), it is possible to obtain a “fingerprint” that can be used to differentiate not only benign from cancerous lesions, but also to differentiate among the different cancerous skin lesions. Figure 8 illustrates that melanoma has large ratios of the 80 Hz, 130 Hz, and 260 Hz peaks, unlike BCC (which has a small 130 Hz peak ratio), SCC (which has a large 50 Hz peak), and AK (which has a very small 260 Hz peak ratio). Using these ratios of the 50/80 Hz peaks and the 80/130 Hz peaks, the different benign and cancerous skin lesions can be “fingerprinted” and used to non-invasively diagnose lesion types prior to surgery (see Table 2). This will assist the dermatologist in assessing the treatment options for difficult-to-diagnose skin lesions, especially in lesions on the face, head, and neck where extensive surgery may be needed.

## 4. Discussion

### 4.1. Fingerprinting Skin Cancers

Using a new non-invasive technique termed VOCT, we have shown that the morphology and stiffness of benign and cancerous lesions are different based on the addition of new cells with increased stiffness (80 Hz), new lesion blood vessels that appear to be less stiff (130 Hz), and fibrous tissue (260 Hz) that is very stiff that are present in all carcinomas. These results suggest that the product of fibrosis that results from cell mutations appears similar for different types of skin cancers, including cancers that have invasive and metastatic potential. Whereas BBCs form primarily fibrotic tissue, other cancers that are invasive and can metastasize also appear to contain large amounts of normal cells (SCC) and large amounts of cancer-associated cells (melanoma). The 3D relationship between the new cells and new blood vessels (80 Hz and 130 Hz) and fibrous tissue (260 Hz) may provide a means for non-invasively assessing the invasive and metastatic potential of a skin cancer.

The ability to noninvasively predict the invasive potential of skin cancers offers the potential for early evaluation of lesions that may be difficult to diagnose. Using VOCT, it is possible to capture and measure the stiffness of lesions as small as 0.05 mm, which is much smaller than can be seen by visual inspection and dermoscopy. Not only do skin cancers contain fibrotic tissue, but they also contain cancer-associated fibroblasts (CAFs) and new friable blood vessels.

#### 4.1.1. Is Fibrotic Tissue Deposition a Sign of Skin Cancer?

It has been reported that an epithelial–mesenchymal transition (EMT) results in a change of the cell phenotype of epidermal cells in epithelial-derived cancers. During EMT, epithelial cells lose cell–cell adhesion attachments and reorganize their cytoskeleton. Cell–cell connections loosen, and cells become motile and resistant to apoptosis [25]. The deposition of large amounts of fibrous tissue is a consequence of tissue breakdown by metalloproteinases and the release of growth factors from altered fibroblasts.

Transforming growth factor beta 1 (TGF-β1), tumor necrosis factor alpha (TNF-α), and hypoxia cooperate in the triggering of EMT both in cancer and fibrosis [25,26], converging in the induction of Snail activity through different mechanisms including the activation of transcription factor NF-kB [26]. In this study, we have shown that all skin cancers are characterized by large amounts of fibrotic tissue deposition; however, there are differences in the cellular and new lesion blood vessel contents of each specific cancer type that may be used as a “fingerprint” of the cancer type and may be related to its potential invasiveness. These studies suggest that the presence of stiff fibrotic tissue may be a sign of a prolonged wound-healing phase associated with cancerous lesion development.

Fibroblasts found in carcinomas appear to exhibit an altered phenotype characterized by the increased expression of α-smooth muscle actin (α-SMA) and fibroblast activation protein (FAP), as well as increased secretion of extracellular matrix proteins (ECM), including fibronectin and fibrous type I collagen [27,28]. CAFs become “activated” in tumors through interactions with cancer cells. CAFs can promote tumor progression to malignancy and metastatic spreading by facilitating underlying processes such as cancer cell migration and invasion [28]. CAFs modify the stromal ECM by enhanced expression and activation of metalloproteinases [29,30]. One of the major pro-metastatic factors derived from CAFs is the cytokine TGF-β [30,31,32,33]. It is possible that all skin cancers have CAFs that promote the deposition of large amounts of fibrous tissue seen in skin lesions. CAFs have been shown to secrete increased levels of TGF-β [30,31,32], which leads to the formation and deposition of dense fibrotic collagen fibers composed mainly of type I collagen (Col I). Col I is another prominent ECM protein with multiple contributions to tumorigenesis [33] and has been reported to promote mammary tumor initiation [34]. The deposition of large amounts of fibrous tissue appears to be a result of the release of TGF-β from altered fibroblasts and may lead to the invasiveness of a tumor.

#### 4.1.2. Invasiveness of Skin Lesions

As cancer progresses, tumor cells obtain the ability to break down the underlying basement membrane and invade into the surrounding tissue. Some cancer cells attain this invasive behavior through the acquisition of a mesenchymal phenotype via EMT [31]. Recent advances in understanding hedgehog signaling in pancreatic fibrogenesis and carcinogenesis highlight new insights into the role of this pathway in fibrosis and cancer progression [34]. The Hippo pathway, which is downstream from the hedgehog pathway, is an evolutionarily conserved signaling pathway with key roles in organ development, epithelial homeostasis, tissue regeneration, wound healing, and immune modulation. Many of these roles are mediated by the transcriptional effectors YAP and TAZ, which direct gene expression via control of the TEAD family of transcription factors [35]. However, both epithelial and endothelial cells can play a role in EMT.

Endothelial cells can be activated by cytokine signaling and express inflammatory markers, which can sustain or exacerbate the inflammatory process. Activated endothelium can recruit and activate leukocytes, thus perpetuating tissue inflammation, and the sustained stimulation of endothelial cells may lead to the endothelial-to-mesenchymal transition that also contributes to fibrosis [36]. Though the fibrosis seen in all skin cancers may be a result of inflammation brought on by CAFs, it is likely that the invasiveness of different cancer types may be driven by the actual cell mutations found in each lesion. The way in which these cancerous mutations drive metastasis may be independent of the fibrosis that is laid down during inflammation. However, the fibrotic plane of crosslinked collagen laid down may serve as a roadmap for cancerous cell migration through the extracellular matrix, which is similar to the role of synthetic collagenous scaffolds that serve as a template for new tissue deposition during the healing of skin and tendinous wounds [37,38].

The products of inflammation may contribute to the formation of stiff fibrous tissue that is seen in the 3D reconstructions reported in this paper to provide a pathway along which cells to migrate. The invasive cancers, SCC, and melanoma, additionally exhibit large numbers of cells (50 Hz and 80 Hz) along with the fibrotic tissue, and melanoma, the most invasive cancer, exhibits large numbers of new cells (80 Hz) and new blood vessels (130 Hz) in addition to large amounts of fibrotic tissue. It is now possible, based on the ratios of the 50, 80, and 130 Hz peaks, to non-invasively determine the lesion composition of cells, new lesion blood vessels, and fibrous tissue to identify potentially metastatic lesions. The ability to classify lesions will prove useful in assisting the dermatologist to identify potentially metastatic epithelial cancers. The “fingerprint” of each cancerous tissue based on the normalized peak height ratios shown in Table 2 is unique and may lend itself to automated analysis using machine learning. If this is possible, then the rapid detection of melanocytic lesions may be possible for small lesions before they are large enough to metastasize.

## 5. Conclusions

Using a new non-invasive technique termed VOCT, the results reported in this pilot study have shown that the morphology and stiffness of benign and cancerous lesions are different based on the addition of new cells with increased stiffness (80 Hz), new lesion blood vessels that appear to be less stiff (130 Hz), and fibrous tissue (260 Hz) present in carcinomas. These results suggest that the product of fibrosis that results from cell mutation appears similar for different types of skin cancers. BCCs have extensive fibrotic tissue deposition, but the number of cells and the amount of new lesion blood vessels is limited. In contrast, cellular proliferation and fibrous tissue deposition are seen to increase in invasive cancers including SCC and melanoma.

## Figures and Tables

**Figure 1 biomolecules-12-01332-f001:**
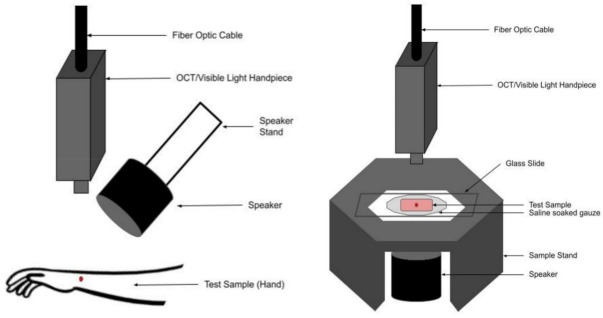
Diagram illustrating the set up for in vivo (**left**) and in vitro (**right**) VOCT measurements made in the reflectance mode.

**Figure 2 biomolecules-12-01332-f002:**
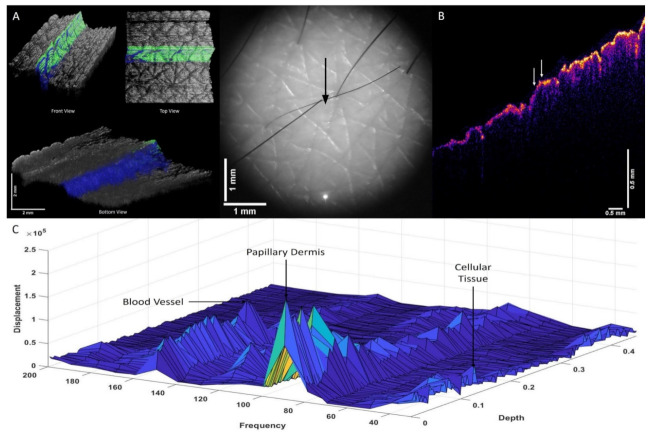
Three-dimensional construction (**A**) of normal skin from a 71-year-old subject illustrating the location of dermal cells (green) and papillary collagen (blue) as well as camera image of the skin. The arrow in the camera image (center) shows the location where the VOCT data were collected, and the OCT image of the skin cross-section is shown in (**B**). (**C**) shows a 3D plot of weighted displacement versus frequency and depth illustrating the location of skin cells, papillary collagen, and blood vessels. Note that the peaks at about 50 Hz (cells), 100 Hz (dermal collagen), and 150 Hz (blood vessels) are found in normal skin. The cells are found primarily in the epidermis and papillary dermis, and the dermal collagen and blood vessels are found in the papillary dermis. The 3D plot of weighted displacement versus frequency versus depth (Figure 2C) was constructed from volume scan data, weighted displacement versus depth measurements, and image pixel intensity versus depth plots.

**Figure 3 biomolecules-12-01332-f003:**
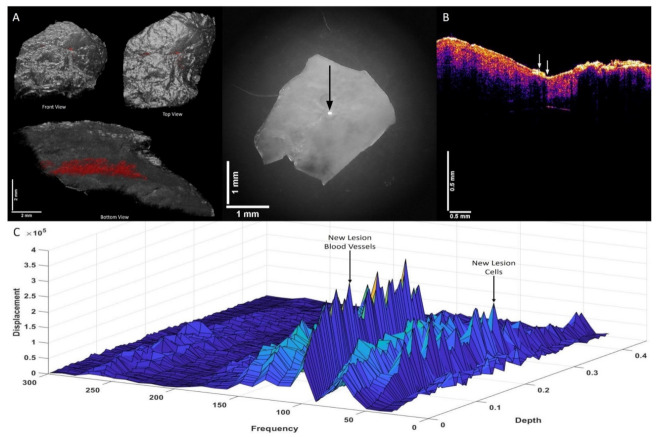
Two-dimensional color-coded OCT images (**A**) of an actinic keratosis (AK) illustrating the location of new dermal cells and new blood vessels in red (left) and a camera image of the lesion (right). The arrow shows the location where the VOCT data were collected. (**B**) shows an OCT image of the cross-section of the point where the data were collected. Note the lesion is seen as a black spot in the area below the arrows. (**C**) shows a 3D plot of weighted displacement versus frequency and depth showing the location of new lesions cells (80 Hz), papillary collagen (100 Hz), and new lesion blood vessels (130 Hz). Note that the peaks at 80 Hz and 130 Hz are not seen in normal skin. The organization of the new cells and new lesion blood vessels are seen in an almost linear fashion (see A) in AK.

**Figure 4 biomolecules-12-01332-f004:**
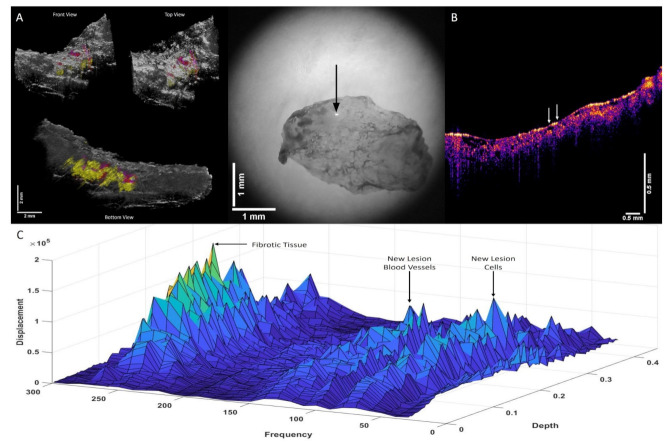
Three-dimensional color-coded OCT images (**A**) (left) of a nodular basal cell carcinoma (BCC), illustrating the location of new cells (80 Hz) and new lesion blood vessels (130 Hz) (red), and fibrotic tissue (260 Hz) (yellow). The camera image of the lesion is shown in the center, and the arrow denotes where the VOCT measurements were made. (**B**) shows an OCT image of the BCC cross-section shown in A. Note the lesion is seen as the small black dot seen below the left arrow. (**C**) shows a 3D plot of weighted displacement versus frequency and versus depth showing the location of skin cells (50 Hz), new cells (80 Hz), dermal collagen (100 Hz,), new blood vessels, (130 Hz) and fibrotic tissue (260 Hz) for a nodular BCC.

**Figure 5 biomolecules-12-01332-f005:**
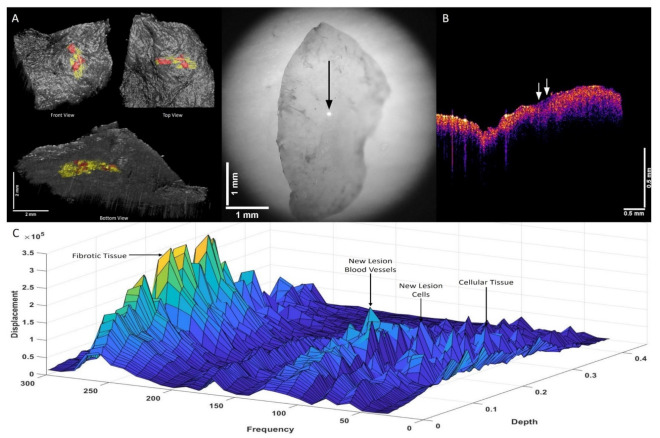
Three-dimensional color-coded OCT images (**A**) of a superficial basal cell carcinoma (BCC) on left and a camera image in center, illustrating the location of new cells (80 Hz) and new lesion blood vessels (130 Hz) (red), and fibrotic tissue (260 Hz) (yellow). The camera image is shown in the center with the arrow pointing to where the VOCT data were collected. (**B**) is an OCT image of a cross-section of superficial BCC and the arrows indicate where the data were collected. (**C**) shows a 3D plot of weighted displacement versus frequency and versus depth, showing the location of skin cells (50 Hz), new cells (80 Hz), dermal collagen (100 Hz), and fibrotic tissue (260 Hz) in a superficial BCC.

**Figure 6 biomolecules-12-01332-f006:**
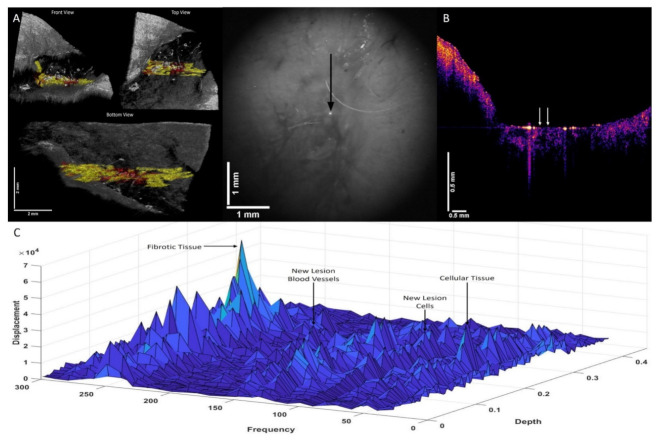
Three-dimensional color-coded OCT images of a squamous cell carcinoma (SCC). (**A**) illustrates the location of new cells (80 Hz) and new lesion blood vessels (130 Hz) (red), and fibrotic tissue (260 Hz) (yellow), as well as a camera image of the lesion showing where the VOCT measurements were made (center). (**B**) shows a cross-section of OCT image of the SCC lesion. (**C**) A 3D plot of weighted displacement versus frequency and depth showing the location of skin cells (50 Hz), new cells (80 Hz), papillary collagen (100 Hz), new lesion blood vessels (130 Hz), and fibrotic tissue (260 Hz). Note that only the peaks at about 50 Hz (cells), 100 Hz (dermal collagen), and 150 Hz (blood vessels) are found in normal skin. Note that the relationship between new cells and fibrotic tissue are found in both islands and linear sheets.

**Figure 7 biomolecules-12-01332-f007:**
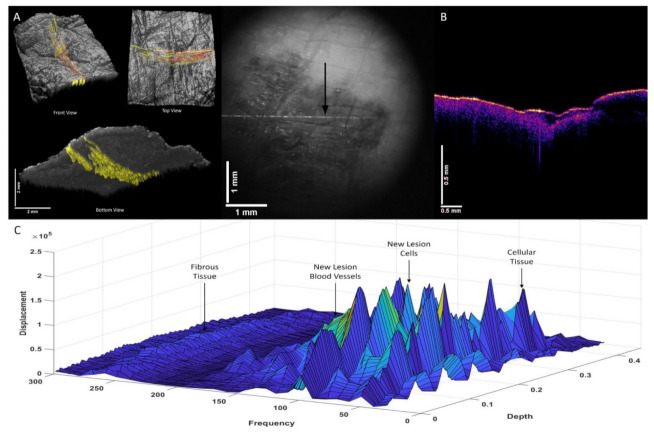
Three-dimensional color-coded OCT images (**A**) of a melanoma illustrating the location of new dermal cells and blood vessels (red) and fibrotic tissue (yellow) and (**B**) an OCT cross-section of a melanoma where the data were collected just below surface of lesion in the center. (**C**) shows 3D plots of weighted displacement versus frequency and depth showing the location of skin cells (50 Hz), new lesion cells (80 Hz), papillary collagen (100 Hz), new lesion blood vessels (130 Hz), and fibrotic tissue (260 Hz). Note that the fibrotic tissue peak (260 Hz) is small in the melanoma due to the absorption of infrared light by the melanin in the lesion.

**Figure 8 biomolecules-12-01332-f008:**
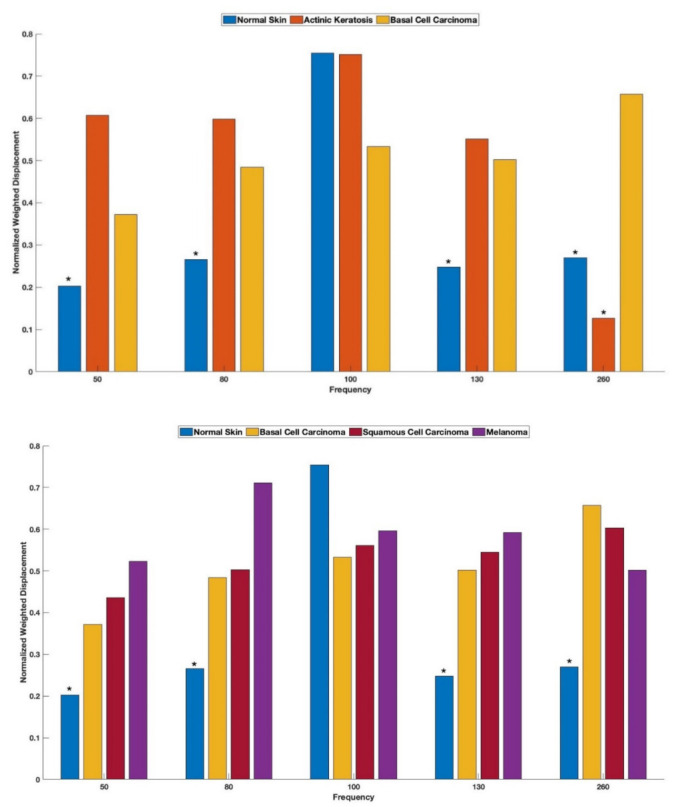
Normalized peak heights obtained from weighted displacement versus frequency plots for skin lesion cross-sections shown in Figure 2B through Figure 7B for 50 Hz, 80 Hz, 100 Hz, 130 Hz, and 260 Hz resonant frequency peaks. The peak heights were normalized by dividing by the largest peak observed in each spectrum for normal skin, AK, BCC, SCC, and melanoma. Note that the nodular and superficial BCC data have been combined in this figure. The top figure illustrates that the normalized peak heights at 50, 80, 130, and 260 Hz for normal skin, AK, and BCC are significantly different at a 0.95 confidence level (*) using a two-tailed Student’s t-test. However, the peak heights were not significantly different than for BCC, SCC and melanoma. The lower figure illustrates that the normalized peak heights were significantly different at a 0.95 confidence level (*) for normal skin at 50, 80, 130, and 260 Hz compared to BCC, SCC and melanoma. The data in this figure illustrate that normal skin has different normalized peak heights from all lesions studied. However, the peak heights, resonant frequencies, and moduli alone are insufficient to determine differences among BCC, SCC, and melanoma. The ratios of normalized peak heights can be used to differentiate among normal skin and different skin cancers, as can be seen in Table 2.

**Table 1 biomolecules-12-01332-t001:** Mean resonant frequency values (top) and standard deviations {} for the resonant frequency peaks found for normal skin, AK, nodular BCC, superficial BCC, SCC, and melanoma. Note that 2 measurements were made at 10 Hz intervals, and as result all resonant frequency peaks do not fall on the same exact frequency.

Resonant Frequency						
Sample	No. of Measurements	50 Hz	80 Hz	100 Hz	130 Hz	260 Hz
Normal Skin	14	50 {0}	NA	100 {0}	NA	NA
AK	7	50 {0}	75.71 {7.68}	104.28 {5.34}	127.14 {4.81}	NA
Nodular BCC	27	52.77 {4.60}	75.55 {5.06}	102.22 {4.23}	125.18 {5.09}	262.59 {4.46}
Superficial BCC	6	54 {5.47}	80 {0}	102 {4.47}	126.67 {5.16}	261.67 {4.08}
SCC	46	49.56 {2.06}	75.86 {4.97}	104.78 {5.05}	127.82 {4.17}	262.82 {4.55}
Melanoma	57	50 {0}	76.31 {4.86}	102.20 {4.18}	128.24 {3.83}	262.98 {4.61}
All values in table are in Hz						
**Modulus**						
**Sample**	**No. of** **Measurements**	**50 Hz**	**80 Hz**	**100 Hz**	**130 Hz**	**260 Hz**
Normal Skin	14	1.231 {0.140}	NA	2.66 {0.32}	NA	NA
AK	7	0.88 {0.160}	1.60 {0.24}	2.54 {0.81}	4.52 {1.206}	NA
Nodular BCC	27	1.467 {0.26}	2.31 {0.41}	3.20 {0.46}	4.41 {0.78}	16.51 {2.65}
Superficial BCC	6	1.525 {0.74}	2.27 {0.40}	2.86 {1.207}	4.46 {0.69}	16.38 {2.82}
SCC	46	1.304 {0.156}	2.02 {0.28}	3.15 {0.45}	4.30 {0.47}	15.68 {1.826}
Melanoma	57	1.459 {0.162}	2.26 {0.29}	3.36 {0.41}	4.81 {0.56}	17.44 {1.938}
All values in table are in MPa						

**Table 2 biomolecules-12-01332-t002:** Normalized ratios of 50/130 Hz and 80/130 Hz peaks for different skin lesions based on Figure 8. Note that BCC, SCC, and melanoma can be differentiated at a 0.95 confidence level (red) based on the peak ratios of 50/80 and 130/80 normalized peak heights using a two tailed Student’s t test.

50 Hz/80 Hz Displacement Ratios		
	SCC	Melanoma
BCC	**0.038**	0.37
Melanoma	**0.031**	
Values in red show statistical significance		
**130 Hz/80 Hz Displacement Ratios**		
	**SCC**	**Melanoma**
BCC	0.4	**0.045**
Melanoma	**0.006**	
Values in red show statistical significance		

## Data Availability

Data contained in this study can be found at optovibronex.com.

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
