# Peer review of "“Fingerprinting” Benign and Cancerous Skin Lesions Using Vibrational Optical Coherence Tomography: Differentiation among Cancerous Lesion Types Based on the Presence of New Cells, Blood Vessels, and Fibrosis"

_biomolecules, 2022, doi:10.3390/biom12101332_

Round 1

Reviewer 1 Report (Previous Reviewer 2)

The authors have addressed all my concerns. I recommend this submission for publication.

Author Response

We thank the reviewer for carefully reading the manuscript and offering helpful comments.

Reviewer 2 Report (New Reviewer)

The manuscript describes the use of vibrational optical coherence tomography (VOCT) for skin lesion assessment. VOCT is a mechano-vibrational spectroscopy method by which the resonant frequencies of tissue can be measured by applying sinusoidal sound waves to the surface and measuring resulting internal displacements. In this work, VOCT is used to analyze biopsied skin lesions and the resulting spectra from VOCT are proposed as a unique “fingerprint” for differentiating between benign and cancerous lesions. Although the proposed method and studies are interesting and have the potential for publication, there is a severe lack of information in the methods used: it is impossible to reproduce the results shown in this manuscript. If this problem along with my other comments below are addressed satisfactorily in a future revision, this work should be in good shape to be publishable.

Major concerns:

  1. There is simply not enough information in this manuscript for replication. It is fine to reference prior work that forms the basis of a publication, but a given publication should be as self-contained as possible, and reliance on other works for description of all methods is not appropriate. In particular, the authors keep referencing a bulk of papers at every turn in the methods (17–24), leaving to the reader the task of sifting through all those papers to discover which one addresses the relevant point, not even mentioning how to solve potentially conflicting methods information across those papers. For instance, I had to spent a great deal of time to determine how the displacements were being measured (there is not mention of phase or Doppler in the present manuscript), and still have no clear idea on how the authors account for motion-induced phase noise. Other basic information is lacking, like the data acquisition strategy (single location?), beam size, speaker power, measurement time. The authors need to include comprehensive methods in this manuscript, and when a complex processing that has been adequately explained in a different publication may be superfluous, they should concisely explain what part of the processing was done according to the methods of which specific paper.

  2. Related to 1., is all this VOCT data acquired in a single spatial lateral location? How was this location chosen? How heterogeneous is the response laterally?

  3. The direct relationship between resonant frequency and stiffness seem to be a very rough approximation that I do not expect it will hold across different experimental setups: the authors argue (in a prior publication) that sound of this frequency range penetrates 8 cm into skin. Wouldn't a lesion in the hand have very different boundary conditions compared to skin in the thigh to make this conversion unreliable? Seems like presence and distance of bone beneath skin would have a significant effect in resonant frequency behavior. In the present study, it seems like the sample holder would play an important role: these biopsies were 250x250µm², a far cry from the boundary conditions in vivo. In particular, the “normal skin” measurement was done in vivo, and thus its use for comparison seems very problematic

  4. The authors state "Maps of the location of the new cells (80Hz peak), new blood vessels (130Hz) and fibrous tissue (280Hz peak) were constructed from measurements of resonant frequency, sample thickness, pixel intensity versus depth scans and volume scans of the tissue." How did they combine all this info? How did they determine where each component of the VOCT spectrum is located? Were they indeed acquiring spectra at different lateral positions? In that case, all the information on how this is done, including the lateral sampling, is missing.

Other comments:

  1. The 3D plots for the VOCT results are poorly formatted and do not benefit interpretation of the results, actually making it more difficult. I recommend these images to be converted into 2D with a specific colomarp. This which would also decrease their footprint on paper and allow the different lesion categories to be displayed next to each other for direct comparison. It would also be easier to overlay dotted lines onto 2D plots for noting the key resonant frequencies instead of labeling them with arrows in 3D and repeating them 6 times in 6 different image captions.

  2. Peak normalization: this is strange, why are all error bars in Fig. 8 the same? I don’t see how this can be, how were they calculated? Also, if I understand the procedure correctly, the peak used for normalization should not have an error bar, it is not clear why it has one.

  3. In Fig. 3 the peak for “new blood vessels” is at 100 Hz, but the text says it should be at 150 Hz.

  4. In Fig. 7, why is only the fibrotic tissue peak affected by absorption and not all other peaks equally?

  5. In all figures: all the resonant frequencies should be labeled with the purported origin, difficult to keep track of all of them.

  6. I can’t make sense of Table 1: what is the meaning of resonant frequency for a resonant frequency? Wouldn't this be maybe amplitude? If the fingerprinting is about how the frequencies of the peak shift (which the table seems to suggest), this is not what is discussed at all in the text. What is the meaning of 0 std? The modulus table should be just the use of Eq. 1 to the values in the first table, but it is clearly not, why? E.g., the 50 Hz column in normal skin and AK is 50 Hz +-0 in both cases, but it translates to two different values with non-zero std in the modulus table.

  7. The authors cite a number of relevant cancer studies in the both the introduction and discussion sections regarding TGF-β, CAFs, and many other things, as well as their respective role in the growth and metastasis of cancer. These paragraphs propose causative biological explanations for the unique “fingerprint” and morphology of different skin lesions, something that is certainly relevant. However, their postulations are mostly weakly connected and would benefit from clearer lead in. For example how does “CAFs modify the stromal ECM by enhanced expression and activation of metalloproteinases” on line 368 relate to VOCT within the context of this study? Given more background research or those with specific prior knowledge, it could probably be inferred, but the authors should really just lay it out more clearly.

  8. In the discussion: In discussion: they have not shown morphology is different, that is not possible with this technique. What the meaning of "3D relationship between new cells and new blood vessels etc."?

  9. The authors say lesions as small as 50 µm can be assessed, but they do not show any such small lesions here.

  10. Last paragraph in discussion repeats what was said at the beginning of the discussion.

  11. The authors talk about "virtual biopsy", but this term has not been defined nor explained.

  12. Conclusions: what is the meaning of "linear geometry"?

  13. Another minor concern regards language and readability, which could be resolved easily with revision. For example, “new blood vessel creation” is more widely referred to as angiogenesis. The discussion section would also benefit greatly from subsections headers, more organization, and more concision in general.

Author Response

Reviewer 3 Report (New Reviewer)

Interesting application of OCT based vibrometry to extract tissue mechanical information based on analysis of the resonant frequency. Authors show evidence that the resonant peaks can be used to provide what they are calling “finger prints” for types of cancerous lesions.  The paper is generally well written and clear.

Minor issues:

Equation 1, the n in f_n should be a subscript.

Line 113, 158: do you mean 55 dB SPL. If you want it be meaningful you should say how far away from the speaker you measured the output. Probably not all that important other than for safety.

Suggestions:

Line 193: Explain the normalization better. Are you measuring the speaker itself to normalize to, if so, where on the speaker? It doesn’t make much sense. It says you described it in a previous publication, but their needs to be a reasonable explanation the reader can follow without having to go back and sift through older papers.

It can be gleaned from previous papers, but it would be useful for the reader if you would briefly describe the physics. 

Author Response

Interesting application of OCT based vibrometry to extract tissue mechanical information based on analysis of the resonant frequency. Authors show evidence that the resonant peaks can be used to provide what they are calling “finger prints” for types of cancerous lesions.  The paper is generally well written and clear.

Minor issues:

Equation 1, the n in f_n should be a subscript

Thank you. Changes have been made. The text and equation have been modified to be identical. The equation format would not allow a subscript so we changed the written text.

Line 113, 158: do you mean 55 dB SPL. If you want it be meaningful you should say how far away from the speaker you measured the output. Probably not all that important other than for safety.

The 55 dB SPL has been added to the revised manuscript. We measured the sound at 2.5 inches from the speaker. The text has been modified to indicate the position of the speaker in the revised text..

Suggestions:

Line 193: Explain the normalization better. Are you measuring the speaker itself to normalize to, if so, where on the speaker? It doesn’t make much sense. It says you described it in a previous publication, but their needs to be a reasonable explanation the reader can follow without having to go back and sift through older papers.

We have added that the normalization process involves measuring the resonant frequency peaks heights of the speaker in the presence of the sample and again in the absence of the sample. The normalized resonant frequency peaks heights are obtained by dividing by the resonant frequency peak heights in the presence of the sample by the peak heights in the absence of the sample.

It can be gleaned from previous papers, but it would be useful for the reader if you would briefly describe the physics

.

We have added that the sample component displacements by the sound wave are inversely related to the modulus of the component being displaced. 

We thank the reviewer for their helpful comments related to the manuscript and now feel the manuscript is ready for publication.

This manuscript is a resubmission of an earlier submission. The following is a list of the peer review reports and author responses from that submission.

Round 1

Reviewer 1 Report

In this manuscript, the authors proposed volumetric visualization of tissue mechanical properties (elastic modulus) of normal and abnormal skin using Optical Coherence Tomography. The novelty only lies in the approach for in vivo application of the method since the method has previously been reported by the authors in vitro. However, the reviewer thinks this should be enough for publication in MDPI Biomolecules. The reviewer would like the authors to address some comments before it can be accepted for publication. 

1. It seems that the study was done in vivo for normal skin and in vitro for abnormal skin (i.e., cancers). Which is not consistent and would not be comparable to some extent? Please explain why the study cannot be done all in vivo?

2. The is no experimental schematic of the study, in vivo and in vitro, I assumed that the in vivo part of the acoustic excitation will be in reflectance mode and the in vitro will be in transmission mode. Schematic will be needed to make the reader understand how the experiment was done.

3. There is not enough explanation about acoustic excitation. Was it the same amplitude and phase for both in vivo and in vitro? What was the value? How the acoustic excitation (e.g., amplitude, distance from tissue, etc.) influences your results?

4. You did not compare your method with the established OCT elastography method, please do this comparison in the introduction part.

5. the new blood vessel can be visualized with your method. Please explain how your method is correlate with the established method like OCT angiography and what is the advantage of your method compares to this method?

6. All the OCT images in the Figures 1, 2, 3, 4, and 5 are not clear in which the images were taken. Please registered with the camera image to show the reader where the exact location of the OCT image.

7. The Figures (B) were taken from one A-scan. Is the arrow on the camera image indicating the location of the A-Scan??

8. Based on table 1(A), I really have a hard time differentiating the tissue types from their resonant frequency. At each acoustic frequency, they have mostly similar resonant frequencies. Please advise and explain more about how you used these values as a “fingerprint”.

Reviewer 2 Report

The authors present a method called vibrational optical coherence tomography to diagnose several skin ailments. This is a big claim and can be impactful; however I am not convinced with the results. Vibrational OCT has been around for some time but other than the current group no other group has shown usefulness of the VOCT in terms of resonant frequency measurement for skin related diagnosis. In the current manuscript also, all the cited references related to VOCT are from their own group.  I have several other concerns which are listed below.

  1. There are several papers from the Ruikang K. Wang group which suggests that the amplitude of the measured displacement in the upper layers depends highly on the properties of lower layers. The reported frequencies in the current manuscript range from 0 to 300 Hz which as per Wang’s work present the properties of deeper layers. The properties of first 400 -500 microns is more represented by frequencies above 1000 Hz range. I am not sure if the authors can diagnose the BCC, SCC etc in the top 400 microns as shown in the manuscript using the frequency range reported in the manuscript. Please see this paper https://doi.org/10.1098/rsif.2011.0583
  2. Previous works from the same group “Shah R, Pierce MC, Silver FH. 2017. A method for nondestructive mechanical testing of tissues andimplants. J Biomed Mater Res Part A 2017” shows that natural frequency in a sample depends highly on the boundary conditions such as length of the sample etc. The same has been observed by other researchers. But in the current manuscript, the authors did not talk about it which make me wonder if the observed resonant frequencies are biased by how the experiment was preformed.
  3. The authors talk about weighted displacement but did not make it clear what it is.
  4. Indicated in “Steven G Adie et al 2009 Phys. Med. Biol. 54 3129” the expected amplitude of the vibrations should be different in different skin layers but I do not see this from figure 1,2, 3, 4, 5 or 6.
  5. In the paper Rakshita Panchal, Luke Horton, Peyman Poozesh, Javad Baqersad, Mohammadreza Nasiriavanaki, "Vibration analysis of healthy skin: toward a noninvasive skin diagnosis methodology," J. Biomed. Opt. 24(1) 015001 (21 January 2019) https://doi.org/10.1117/1.JBO.24.1.015001, The natural frequency of normal skin measured using DIC is around 140 to 156 Hz. The authors claim several frequencies observed below this range which were not seen in previous works which again raise the question if the observed natural frequencies are biased by the experimental procedure.
  6. A minor comment; the statement in the manuscript that “Both cancer cells and cancerous tissue are stiffer than normal tissue” is only correct in few cancer types.

Author Response

please see attacehd

Round 2

Reviewer 2 Report

I have big confusion about the term ‘transverse deflection’. In OCT community, the displacement along the beam is called axial or longitudinal displacement. In my understanding, transverse deflection means perpendicular to the beam but what I understood from previous papers from the group is that the measured displacement is along the beam. Can authors clarify this. If it is indeed along the beam than its hard for me to believe that the lower layers of skin have no effect on the measurements in the frequency range 0-300 Hz. But since there are no past studies related to this, I guess I can accept the claims from the authors.